# Receptor-Binding Domain (RBD) Antibodies Contribute More to SARS-CoV-2 Neutralization When Target Cells Express High Levels of ACE2

**DOI:** 10.3390/v14092061

**Published:** 2022-09-16

**Authors:** Ariana Ghez Farrell, Bernadeta Dadonaite, Allison J. Greaney, Rachel Eguia, Andrea N. Loes, Nicholas M. Franko, Jennifer Logue, Juan Manuel Carreño, Anass Abbad, Helen Y. Chu, Kenneth A. Matreyek, Jesse D. Bloom

**Affiliations:** 1Basic Sciences Division and Computational Biology Program, Fred Hutchinson Cancer Center, Seattle, WA 98109, USA; 2Department of Genome Sciences & Medical Scientist Training Program, University of Washington, Seattle, WA 98195, USA; 3Howard Hughes Medical Institute, Seattle, WA 98195, USA; 4Division of Allergy and Infectious Diseases, University of Washington, Seattle, WA 98109, USA; 5Department of Microbiology, Icahn School of Medicine at Mount Sinai, New York, NY 10029, USA; 6Department of Pathology, School of Medicine, Case Western Reserve University, Cleveland, OH 44106, USA

**Keywords:** SARS-CoV-2, neutralization assay, pseudovirus, ACE2, target cell receptor expression, RBD, NTD, neutralizing antibody epitopes

## Abstract

Neutralization assays are experimental surrogates for the effectiveness of infection- or vaccine-elicited polyclonal antibodies and therapeutic monoclonal antibodies targeting SARS-CoV-2. However, the measured neutralization can depend on the details of the experimental assay. Here, we systematically assess how ACE2 expression in target cells affects neutralization by antibodies to different spike epitopes in lentivirus pseudovirus neutralization assays. For high ACE2-expressing target cells, receptor-binding domain (RBD) antibodies account for nearly all neutralizing activity in polyclonal human sera. However, for lower ACE2-expressing target cells, antibodies targeting regions outside the RBD make a larger (although still modest) contribution to serum neutralization. These serum-level results are mirrored for monoclonal antibodies: N-terminal domain (NTD) antibodies and RBD antibodies that do not compete for ACE2 binding incompletely neutralize on high ACE2-expressing target cells, but completely neutralize on cells with lower ACE2 expression. Our results show that the ACE2 expression level in the target cells is an important experimental variable, and that high ACE2 expression emphasizes the role of a subset of RBD-directed antibodies.

## 1. Introduction

Neutralization assays are the most widely used experimental method to assess immunity elicited by SARS-CoV-2 vaccination and infection. However, the neutralization measured in the lab depends on the details of the assay. Different assays use both different viral systems (live virus versus pseudovirus) [1,2,3,4] and different target cells (cells engineered to overexpress ACE2 versus cells that endogenously express ACE2, such as Vero) [5,6,7]. Previous studies have shown that some antibodies can have markedly different neutralizing activities depending on which viral systems [1,3,8,9,10] or target cells [5,6,7,11] are used.

Here, we systematically assess how target-cell ACE2 expression impacts the contribution of different types of antibodies to the neutralizing activity of polyclonal serum as measured in lentiviral pseudotype assays. We show that RBD-targeting antibodies contribute more to serum neutralization when the target cells express more ACE2. We also show that individual monoclonal antibodies targeting epitopes outside the receptor-binding motif of the RBD have much poorer neutralization on high versus low ACE2-expressing target cells. Overall, our work demonstrates that ACE2 expression in target cells dramatically influences the measured neutralization by some antibodies, and that RBD-targeting antibodies contribute comparatively more to neutralization in high ACE2-expressing target cells.

## 2. Materials and Methods

### 2.1. Generation of 293T Cells Engineered to Express Different Levels of ACE2

We used previously described 293T-based landing pad cells to create cell clones that express different levels of ACE2 by modifying the Kozak sequence controlling the translation of the gene [12]. Prior to modification, these HEK 293T LLP-Int-BFP-IRES-iCasp9-Blast clone 3 landing pad cells [12] were maintained in D10 growth media (Dulbecco’s Modified Eagle Medium supplemented with 10% heat-inactivated fetal bovine serum, 100 U/mL penicillin, 100 µg/mL streptomycin, and 2 mM L-glutamine) supplemented with 2 μg/mL doxycycline and 10 µg/mL blasticidin. These landing pad cells were modified with ACE2 transgenic sequences by transfecting 600,000 cells with 1200 ng of Kozak-variable AttB_ACE2-miRFP670_IRES_mCherry-H2A-P2A-PuroR recombination plasmid mixed with 5 μL of Fugene 6 reagent per six-well. The Kozak sequences preceding ACE2 were GCCACCATG, TATCTAATG, TATTTCATG, and AATTTTATG corresponding to “high”, “medium”, “low”, and “very low” cells, respectively [13]. The cells were transfected in D10 + doxycycline growth media until day 3 after transfection, when AP1903 (ApexBio, B4168, Boston, MA, USA) was added to a final concentration of 10 nM to kill off unmodified landing pad cells. Once the cells reached ~ 10% confluence, the cells were switched to D10 + doxycycline growth media containing 1 μg/mL puromycin to achieve a roughly pure population of ACE2 transgenic cells.

### 2.2. Cell Lines

293T clones expressing different levels of ACE2 (described above) were grown in D10 growth media supplemented with 2 μg/mL doxycycline, which is required to induce their ACE2 expression from a landing pad. After several passages, 0.75 μg/mL of puromycin was added to the cell media to remove cells no longer expressing the transgenic locus.

For Figure 1A, we used Vero E6 (American Type Culture Collection, CRL-1586, Manassas, VA, USA) to look at endogenous ACE2 expression in these cells, and 293T cells (American Type Culture Collection, CRL-3216) for the “no ACE2” control. For Appendix A, the 293T-ACE2 cells are those previously described in [14] (available at Biodefense and Emerging Infectious Research Resources Repository (BEI Resources), NR-52511, Manassas, VA, USA) and the 293T-ACE2-TMPRSS2 cells are those described in [15] and were a gift from Carol Weiss. Vero E6, 293T, 293T-ACE2, and 293T-ACE2-TMPRSS2 cells were maintained in standard D10 growth media.

### 2.3. Flow Cytometry Analysis for ACE2 Expression

Cells were plated at 5 × 10^5^ cells per well in a six-well plate. For 293T clones expressing different levels of ACE2, 2 μg/mL of doxycycline was added to cell media during plating. The next day cells were trypsinized and centrifuged at 300× *g* for 4 min. After washing with 1 mL of FACS buffer (phosphate-buffered saline (PBS) + 2% bovine serum albumin (BSA)), cells were resuspended in 500 μL of rabbit anti-ACE2 antibody (Abcam, ab272500) at 1 µg/mL. After 1 h incubation on a rotator at 4 °C, the cells were washed with 1 mL of FACS buffer and resuspended in 500 μL of goat anti-rabbit Alexa Fluor 488 antibody (Abcam, ab150077) at 0.67 μg/mL. After 1 h incubation on a rotator at 4 °C, the cells were washed twice and resuspended in 500 μL of FACS buffer. The cells were analyzed by flow cytometry using the BD LSRFortessa X-50 cytometer and data were plotted using FlowJo software (Version 10, BD Biosciences, Ashland, OR, USA). Geometric mean fluorescence values were used to determine the ACE2 expression relative to “high” clone.

### 2.4. Generation of Spike-Pseudotyped Lentiviral Particles

Lentiviral pseudotyping was performed as previously described in [14]. To generate pseudoviruses, the following plasmids were used: codon-optimized Wuhan-Hu-1 spike expression plasmid containing D614G mutation and 21 amino acid deletion in the C-terminal domain (sequence and plasmid available from BEI Resources, NR-53765); lentiviral helper plasmid-encoding Gag/Pol (NR-52517); pHAGE6_Luc_IRES_ZsGreen plasmid-encoding luciferase and ZsGreen reporter genes in a lentiviral backbone.

To generate spike-pseudotyped lentiviral particles, 293T cells were seeded at 2.5 × 10^6^ cells per 10 cm dish. Then, 16–24 h later, the cells were transfected using 15 µL of BioT transfection reagent (Bioland Scientific, Paramount, CA, USA) with 5.7 µg of a lentiviral backbone plasmid, 2.6 µg of Gag/Pol helper plasmid, and 1.7 µg of spike expression plasmid. At 24 h post transfection, the cell culture media was replaced with fresh D10 media. At ~60 h post transfection, the virus in the cell media was harvested by passing cell supernatant through a surfactant-free cellulose acetate 0.45 µm syringe filter (Corning, 431220, Union City, CA, USA), and stored at −80 °C.

### 2.5. Titering of Spike-Pseudotyped Lentiviral Particles

Very low, low, medium, and high ACE2 clones were seeded onto a poly-L-lysine-coated black-walled 96-well plate at 1.25 × 10^4^ cells per well in 50 µL D10 growth media, supplemented with 2 µg/mL doxycycline and 2.5 µg/mL of amphotericin B. The next day, 100 µL of serially diluted virus was added to each well. At ~48–50 h post infection, luciferase activity was measured using the Bright-Glo Luciferase Assay System (Promega, E2620, Madison, WI, USA). The bottom of each plate was covered with a black bottom sticker (Thermo Fisher Scientific, NC9425162, Waltham, MA, USA) and luciferase activity was read using a Tecan infinite M1000Pro plate reader (Tecan, Männedorf, Switzerland). Dilution series were used to calculate virus titers in relative luciferase units (RLU) per µL.

### 2.6. Human Sera

Serum samples were collected with informed, written consent as part of the prospective longitudinal Hospitalized or Ambulatory Adults with Respiratory Viral Infections (HAARVI) cohort of individuals with SARS-CoV-2 infection in Seattle, WA. The samples were collected between January and June 2021. The ten serum samples used in this study come from individuals who were infected with SARS-CoV-2 between March and June of 2020 and were subsequently vaccinated. The samples were from seven females and three males, of age range 36–72 years, and a mean age of 50.3 years (Appendix A). The samples were collected an average of 390 days (range 296–454 days) post symptom onset and an average of 18 days (range 8–36) post second dose of a primary mRNA vaccine series. All sera were heat-inactivated at 56 °C for 60 min prior to storage at −80 °C.

### 2.7. Depletion of RBD-Binding Antibodies from Sera

SARS-CoV-2 Wuhan-Hu-1 RBD-coupled magnetic beads (AcroBiosystems Inc., MBS-K002, Newark, DE, USA) were reconstituted at 1 mg/mL in assay buffer (PBS + 0.05% BSA) and washed three times with fresh assay buffer, each time maintaining 1 mg/mL concentration. The washed beads were mixed with sera at a ratio of 3 parts beads to 1 part sera and incubated with end to end rotation for 2 h at room temperature or at 4 °C overnight. After incubation, a magnet was used to separate the beads (along with RBD-binding antibodies) from the supernatant (containing the non-RBD-binding antibodies) and the supernatant was transferred to a tube of fresh beads for another round of depletion. A total of four rounds of depletion were performed for each serum sample and beads were each time applied at a ratio of 3 parts beads to 1 part serum. An aliquot of serum after each round of depletion was reserved for enzyme-linked immunosorbent assay (ELISA) and kept at 4 °C. Non-depleted sera were diluted to the same degree as the final dilution of depleted sera (1:4) in PBS + 0.05% BSA, dictated by the 3:1 beads-to-sera ratio.

### 2.8. Measurement of Sera Binding to RBD by Enzyme-Linked Immunosorbent Assay (ELISA)

ELISA experiments for sera RBD-binding were performed as previously described in [16]. Immunlon 2HB 96-well plates (Thermo Scientific 3455, St Louis, MO, USA) were coated with 50 µL of Wuhan-Hu-1 RBD protein (gifted from the Institute for Protein Design) at 0.5 µg/mL in PBS and stored at 4 °C overnight. The next day, the plates were washed three times with PBS containing 0.1% Tween 20 (PBS-T) using a plate washer (Tecan HydroFlex) and then 200 µL of blocking buffer (PBS-T containing 3% nonfat dry milk) was added to each well. The plates were incubated at room temperature for 1 h. In a separate plate, RBD-depleted and non-depleted sera were serially diluted (starting at 1:100 dilution) in PBS-T containing 1% nonfat dry milk. Blocking buffer from the RBD-covered plate was removed and 100 μL of diluted serum was added to each well. After a 2 h incubation at room temperature, the plates were washed three times using a plate washer, and 50 µL of Human IgG-Fc Fragment Antibody (Bethyl Laboratories, A80-104P, Montgomery, TX, USA) at 0.33 µg/mL was added to each well. The plates were incubated at room temperature for 1 h and then washed three times using a plate washer. Following the wash, 100 µL of TMB/E horseradish peroxidase (HRP) substrate (Millipore Sigma, ES001, Burlington, MA, USA) was added to each well. After a 5 min incubation, the reaction was stopped by adding 100 µL of 1 N HCl per well. OD_450_ values were read on a Tecan infinite M1000Pro plate reader.

### 2.9. Spike-Pseudotyped Lentivirus Neutralization Assays

Very low, low, medium, and high clones were seeded at 2.5 × 10^4^ cells per well onto poly-L-lysine-coated black-walled 96-walled plates in 50 µL D10 media supplemented with 2 µg/mL of doxycycline and 2.5 µg/mL amphotericin B. The next day, the depleted and non-depleted sera were serially diluted (starting with a 1:25 dilution) and mixed with pseudovirus at a 1:1 ratio. The virus–sera mixes were incubated for 1 h at 37 °C and then 100 µL was added to the pre-seeded plates. The same virus stock and dilution was always used for matched depleted and non-depleted sera across all four cell clones (targeting > 500 k RLUs per well). Each sample was run in duplicate, and each row contained two inoculated no-serum positive controls and one no-infection negative control. After 48–50 h incubation, luciferase activity was measured using the Bright-Glo Luciferase Assay System (Promega E2610). A black sticker was applied to the bottom of each plate to reduce background and luciferase activity was measured using the Tecan infinite M1000Pro plate reader.

Fraction infectivity was calculated for each serum-containing well by subtracting the background signal from the negative control and normalizing by the average of two positive control wells in the same row. Inhibitory concentration 50% (IC_50_) and its reciprocal, neutralization titer 50% (NT_50_) were calculated using the neutcurve software package (https://jbloomlab.github.io/neutcurve/, (accessed on 20 May 2022), version 0.5.7), by fitting a Hill curve and fixing the top of the curve to one and the bottom to zero.

For neutralization assays using monoclonal antibodies, the following modifications to the above protocol were made: starting dilution for Ly-CoV555 was 0.667 µg/mL, for S309 was 6 µg/mL, and for 4A8 was 1 µg/mL; IC_50_ values were calculated by fixing the top of the neutralization curve to one and leaving the bottom unfixed.

## 3. Results

### 3.1. 293T Target Cells Expressing Different Amounts of ACE2 Protein

To investigate the effect of target-cell ACE2 expression on SARS-CoV-2 neutralization, we exploited a previously described method for creating HEK 293T cells with defined ACE2 protein expression levels [13]. Briefly, this method involves integrating a single copy of the ACE2 gene into an engineered locus in the 293T cell’s genome, and modulating the protein expression level by altering the Kozak sequence [13]. Using this approach, we generated 293T clones expressing very low, low, medium, and high levels of ACE2 (Figure 1A). The level of ACE2 expression across these clones spans a range of ~30 fold. The “high” 293T cell clone is most similar in ACE2 expression to previously described 293T-ACE2 cells [14] commonly used in lentiviral pseudotype neutralization assays (Appendix A). We also compared the ACE2 expression in the 293T clones to Vero E6 cells, which are commonly used for VSV pseudotyped assays and live-virus neutralization assays [4,6]. Vero E6 cells have a wider range of ACE2 expression than our ACE2-expressing 293T clones, but are most similar in mean expression to the “very low” 293T clone (Figure 1A).

As expected, the infectability of the 293T clones by spike-pseudotyped lentiviral particles paralleled their ACE2 expression (Figure 1B), with the “very low” clone (which expressed ~30-fold less ACE2) being ~8-fold less infectable.

### 3.2. Target-Cell ACE2 Expression Affects the Contribution of RBD-Targeting Antibodies to Neutralization by Polyclonal Serum

Prior studies using high ACE2-expressing cells have suggested that RBD-targeting antibodies are responsible for the majority of the neutralizing activity in polyclonal human sera [17,18,19,20,21]). To assess whether the importance of RBD-targeting antibodies for neutralization depends on target-cell ACE2 expression, we depleted RBD-targeting antibodies from polyclonal sera (Figure 2A and Appendix A) from human individuals who were infected with SARS-CoV-2 in early 2020 and then vaccinated with either Pfizer or Moderna mRNA vaccines in early 2021. We then performed spike-pseudotyped (Wuhan-Hu-1 with D614G) lentiviral neutralization assays on all four 293T cell clones with different ACE2 expression levels using RBD-depleted and non-depleted sera.

RBD-targeting antibodies were responsible for a larger share of the overall serum neutralizing activity when the target cells expressed high levels of ACE2 (Figure 2B–D). For instance, RBD-targeting antibodies contributed ~99% of the neutralizing activity measured using the high ACE2 cells, but only ~90% of the neutralizing activity measured using the low and very-low ACE2 cells (Figure 2B–D). These results show that while RBD-targeting antibodies provide the majority of neutralization measured at all target-cell ACE2 expression levels, their relative contribution is larger in high ACE2 cells (where they contribute almost all the activity) than low ACE2 cells (where non-RBD antibodies make an appreciable minority contribution to the overall neutralizing activity) (Figure 2D).

There was also a modest trend for the overall measured serum neutralization titers to be greater in cells that expressed lower levels of ACE2 (Figure 2B). This was true for both the non-depleted and RBD-depleted sera, but the trend was much stronger for the RBD-depleted sera. In particular, RBD-depleted sera always retained measurable neutralizing activity in the very low, low, and medium ACE2 target cells, but some sera lost all detectable neutralization in the high ACE2 target cells (Figure 2B).

### 3.3. Monoclonal Antibodies to Epitopes outside the RBD’s Receptor-Binding Motif Are Much Less Potent on High ACE2 Target Cells

To better understand why the RBD-targeting portion of polyclonal serum makes a larger contribution to neutralization in high ACE2 target cells, we examined neutralization by monoclonal antibodies targeting distinct epitopes in spike. We selected three monoclonal antibodies: Ly-CoV555 binds to the receptor-binding motif of the RBD and competes with ACE2 binding, S309 binds the RBD outside receptor-binding motif and does not compete with ACE2 binding, and 4A8 binds the N-terminal domain (NTD).

We found that antibodies that bound outside the receptor-binding motif were much less potent on target cells that expressed high levels of ACE2 (Figure 3). Specifically, the ACE2-competing antibody that binds the receptor-binding motif (Ly-CoV555) was only modestly affected by the target-cell ACE2 expression level, achieving full neutralization on all target cells with a ~5-fold reduction in inhibitory concentration 50% (IC_50_) on the high ACE2 target cells relative to the very low ACE2 cells. In stark contrast, the neutralization by the two antibodies that bound outside the receptor-binding motif was dramatically impaired on high ACE2 target cells. Neither the NTD-targeting antibody 4A8 nor ACE2-non-competing RBD antibody S309 achieved full neutralization on the high ACE2 cells even at a very high concentration. However, both antibodies showed much better neutralization in lower ACE2 cells, with S309 achieving full neutralization and 4A8 reaching a plateau of ~90% neutralization (as opposed to ~50% plateau on high ACE2 cells) (Figure 3). Overall, these monoclonal antibody results suggest a mechanistic explanation for the serum results described above: antibodies that target epitopes outside the RBD’s receptor-binding motif are much less potent on high ACE2 target cells.

## 4. Discussion

Our results show that RBD-directed antibodies make a greater contribution to viral neutralization when the target cells express high levels of ACE2. This finding was consistent over ten different human sera: on high ACE2-expressing target cells, RBD-directed antibodies were consistently responsible for ~99% of the neutralizing activity of human sera, whereas on lower level ACE2-expressing cells, their activity dropped to ~80–95%. So, while our results corroborate prior studies showing that RBD antibodies are the dominant contributors to serum neutralization [17,18,19,20,21], they also show that the magnitude of this dominance depends on the ACE2 expression level of the target cell.

Our experiments with monoclonal antibodies provide some insight into why RBD-directed antibodies contribute more to serum neutralization in cells with high ACE2 expression. Neutralization by an RBD-targeting antibody that directly competes with ACE2 for RBD binding is only modestly affected by the target-cell ACE2 expression level. However, both an RBD-targeting antibody that targets a non-ACE2 competing epitope and an NTD-targeting antibody neutralize much more poorly on cells with high ACE2 expression; similar results have been reported by other studies [5,6,7]. Therefore, in the context of polyclonal serum, the relative neutralization contributions of antibodies targeting different epitopes shifts with target cell ACE2 expression. Why the neutralization potency of different antibodies differs based on ACE2 expression remains unclear, in part because the neutralization mechanisms of non-ACE2-competing antibodies are still incompletely understood [1,22,23,24,25,26]. We also note that there is literature suggesting that the neutralization of other viruses by antibodies and drugs sometimes depends on target-cell receptor expression [27,28].

Our study is unable to definitively answer the most important question it raises: What target cell ACE2 expression provides the most biologically relevant measure of SARS-CoV-2 neutralization? Serum-neutralizing antibody titers are a correlate of protection for SARS-CoV-2 [29,30,31], meaning neutralization titers measured in the lab correlate with protection in humans, although the exact levels of neutralizing antibodies associated with protection are not defined. However, laboratory work alone cannot determine which target cells provide experimentally measured titers most predictive of human protection—although we do note that the airway cells infected during typical human cases express fairly low levels of ACE2 (potentially even lower than Vero cells) [32,33,34]. The possibility that lower ACE2-expressing target cells may be more biologically relevant is also supported by the observation that the monoclonal antibody S309 (sotrovimab) provides strong protection in humans despite achieving potent neutralization only when measured in low or moderate ACE2-expression target cells [1,6,35].

A limitation of our study is that we only measured neutralization with spike-pseudotyped lentiviral particles and did not perform multi-cycle neutralization assays with authentic SARS-CoV-2 [6,36,37] or spike-expressing chimeric VSV [2,4,11,20]. However, some prior studies using authentic SARS-CoV-2 have suggested that neutralization by non-ACE2-competing RBD antibodies is more potent on cells expressing lower ACE2 [6,7], suggesting the trends we observe likely extend beyond spike-pseudotyped lentiviral particles.

The most important implication of our work is that target-cell ACE2 expression is an important experimental variable for SARS-CoV-2 neutralization assays. For instance, some new SARS-CoV-2 vaccine candidates attempt to elicit higher levels of antibodies to more conserved non-ACE2-competing RBD epitopes [38] or non-RBD regions of the spike [39]. The measured neutralization titers elicited by such vaccine candidates are likely to depend to some extent on target-cell ACE2 expression. A similar dependence is likely when assessing the relative neutralization of different SARS-CoV-2 variants of concern, which often have mutations both within and outside the ACE2-binding motif of the RBD. Careful reporting of ACE2 expression by the target cells used to assess neutralization titers will therefore aid in interpreting and comparing neutralization studies of different vaccine candidates and viral variants.

## Figures and Tables

**Figure 1 viruses-14-02061-f001:**
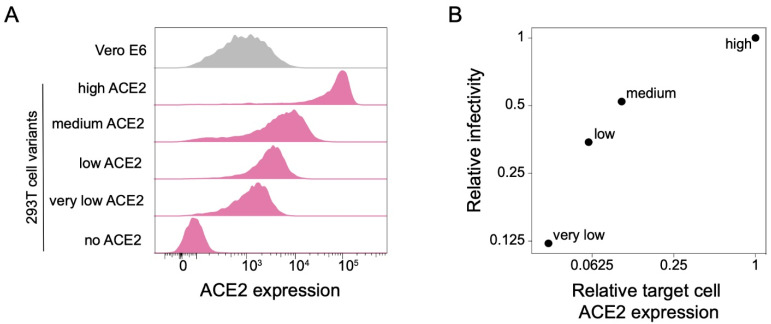
293T cell clones expressing ACE2 at different levels. (**A**) ACE2 expression in 293T cells engineered to express different levels of ACE2. ACE2 surface expression was measured by flow cytometry, and the histograms show the distribution of expression levels over a population of cells. Vero E6 cells are included for comparison. (**B**) Relationship between ACE2 expression in the four 293T target cell clones and infection by lentiviral particles pseudotyped with the SARS-CoV-2 D614G spike.

**Figure 2 viruses-14-02061-f002:**
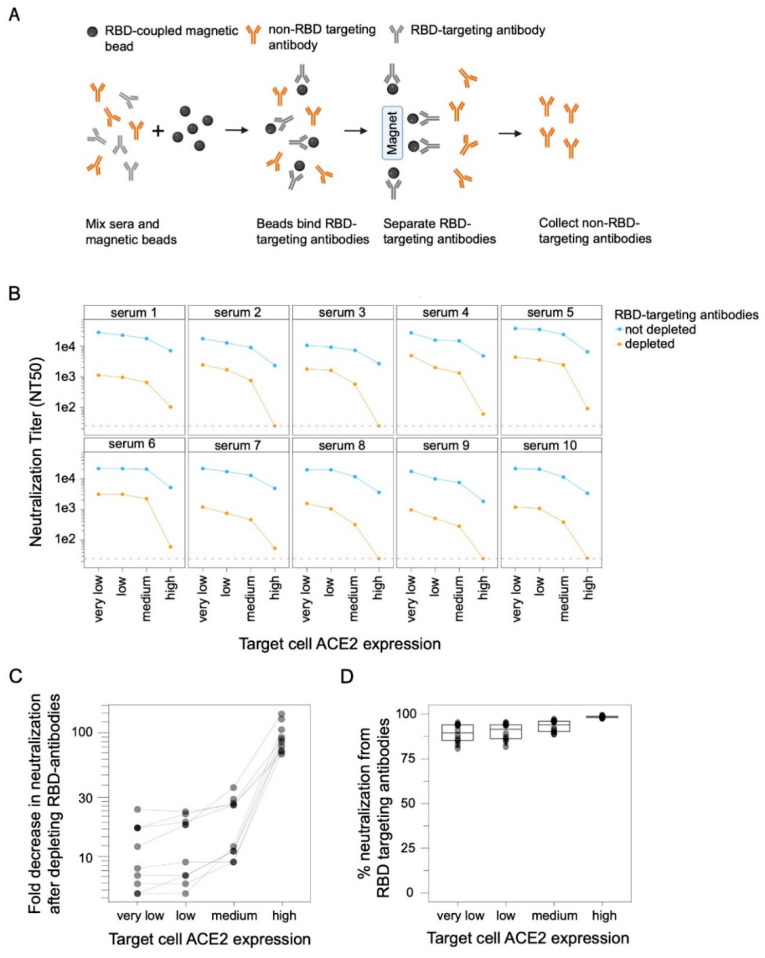
RBD-targeted antibodies make a larger contribution to serum neutralization when target cells express higher levels of ACE2. (**A**) Process for depleting RBD-targeted antibodies from polyclonal human serum. RBD-coupled magnetic beads are incubated with sera. The RBD-targeting antibodies bind the beads and are then removed from sera by magnetic separation. RBD-targeting antibody depletion was confirmed by ELISA (Appendix A). This figure was created with BioRender.com. (**B**) Neutralization of D614G spike-pseudotyped lentiviral particles by polyclonal human sera from ten different individuals with or without depletion of RBD-targeting antibodies, as measured on target cells expressing different levels of ACE2. Neutralization is reported as the neutralization titer 50% (NT_50_), which is the reciprocal serum dilution that neutralizes half of the virus. The dashed red line represents the limit of detection (NT_50_ of 25), and values less than this limit are assigned a value of 25. (**C**) Fold decrease in neutralization after depleting the RBD antibodies and (**D**) percent of neutralization due to RBD-targeting antibodies for the sera shown in panel (**B**), calculated by subtracting NT_50_ values for depleted sera from non-depleted sera and expressed as percentage of non-depleted sera neutralization.

**Figure 3 viruses-14-02061-f003:**
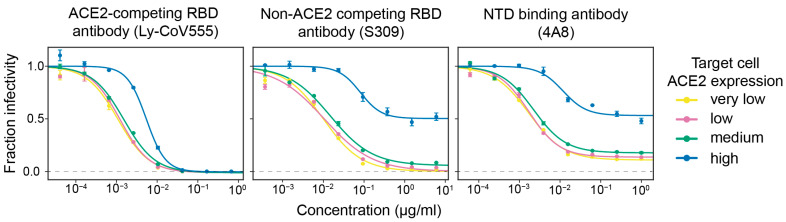
High ACE2 expression in target cells strongly reduces neutralization by monoclonal antibodies that bind spike epitopes outside the receptor-binding motif. Neutralization curves for three monoclonal antibodies that target different epitopes: Ly-CoV555 binds the RBD’s receptor-binding motif and competes with ACE2 binding, S309 binds a RBD epitope outside the receptor-binding motif, and 4A8 binds the NTD. The dashed gray line indicates zero infectivity. Neutralization assays were performed using D614G spike-pseudotyped lentiviral particles.

## Data Availability

The code used to analyze and plot ACE2 expression, virus titers, and virus neutralization is available at https://github.com/jbloomlab/Ace2_expression_neuts, (accessed on 16 August 2022).

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
