# Peer review of "Receptor-Binding Domain (RBD) Antibodies Contribute More to SARS-CoV-2 Neutralization When Target Cells Express High Levels of ACE2"

_viruses, 2022, doi:10.3390/v14092061_

Round 1
Reviewer 1 Report
This paper reports developing different cell lines expressing different levels of ACE2 (very low, low, medium, and high) to evaluate the sarbecovirus pseudovirus-based assays that are important for SARS-CoV-2 research. A strength of the paper is that they depleted RBD targeting antibodies from convalescent and vaccinated serum and evaluated undepleted and depleted sera for neutralization of the D614G pseudovirus using the 4 different cell lines. They report the following conclusions:
· RBD targeting antibodies contribute more to neutralization on high ACE2 expressing cell lines, less to low ACE2 expressing cell lines.
· RBD depleted antibodies (after RBD depletion) showed low to no neutralization in high ACE2 cell lines.
· Neutralization by non-ACE2 competing antibodies were dramatically affected by the high ACE2 cell lines, where they didn’t completely neutralize. An example is the class 3 antibody, S309 (Sotrovimab).
This work is important in addressing the differences researchers see in neutralization with the same antibodies, as well as differences in neutralization elicited by similar or different vaccine platforms.
Suggestions for including in the paper as discussion points:
· Is the level of ACE2 expression on human airway epithelial cells known? If so, which ACE2 cell line does it resemble the most? Knowing this would be highly relevant to using neutralization results to predict efficacy.
· It would also be interesting to see if these trends hold for other sarbecovirses such as SARS-CoV and for variants such as Omicron BA.5.
Author Response
Thanks for the helpful and very favorable comments. To address the two specific points:
- ACE2 expression in human airway cells is known to be very low, and although it's hard to quantitatively compare to Vero cells it may be even lower. We have added text to the Discussion along with citations addressing this point.
- Although we agree it would be interesting to repeat our study with other viral variants, this is beyond the scope of our current work.
Reviewer 2 Report
Summary
Farrell et al. report findings on how ACE2 expression levels affect antibody neutralization of SARS-CoV-2. Using lentiviral pseudoviruses bearing the D614G spike and 293T cells expressing various levels of surface ACE2, they test ten COVID-19 convalescent serum samples for neutralization activity. They deplete serum of RBD-binding antibodies by multiple rounds of recombinant RBD adsorptions and show that the neutralization activity is greatly reduced for most serum samples in cells with high ACE2 levels, whereas neutralization activity is much less affected in cells with low and very low levels of ACE2 expression. To explore the regions of spike accounting for the findings, they assessed one monoclonal each that targets an epitope in the receptor binding motif (RBM), outside the RBM, and the N-terminal domain. They found that the antibodies targeting regions outside of the RBM were less potent on cells with high ACE2 levels. The authors conclude that RBD-directed antibodies make a greater contribution to neutralization on cells with high levels of ACE2 than cells with low levels of ACE2.
Comment
The report confirms and extends prior reports showing that ACE2 receptor levels can affect neutralization of SARS-CoV-2 spike-lentiviral pseudoviruses and that RBD-directed antibodies are the predominant neutralizing antibodies in COVID-19 convalescent sera. Careful ACE2 level titrations and depletion experiments add to existing literature, which is adequately referenced. Experiments appear to be done well, and the manuscript is clearly written. The report highlights assay effects. Potential mechanisms behind the findings in remain unclear.
Minor comment
· Authors may wish to comment on interpretation of NT50 titers when neutralization curves do not reach complete neutralization compared to NT50 titers based on curves that do reach complete neutralization.
Author Response
Thanks for the favorable comments. We agree the mechanism remains unclear and is an interesting topic for future work.
In regards to the point about incomplete neutralization, this is a good observation. In our study, we do not report NT50s for any of the antibodies that plateau at less then complete neutralization to avoid this issue. For serum that do not completely neutralize it is simply due to the curve not ever going down to a plateau, so we report those NT50s as upper bounds.
Reviewer 3 Report
In this manuscript, Farrell, Dadonaite, et al. evaluated how the expression level of ACE2 receptor may affect the experimental results in pseudovirus neutralization assays. The authors constructed target cells with different ACE2 expression levels, and tested polyclonal human serum samples with or without depletion of anti-RBD antibodies, which showed anti-RBD antibodies played a bigger role in neutralizing ability in high ACE2-expressing target cells, while antibodies targeting other epitopes were responsible for a larger share of neutralization in low ACE2-expressing ones. Neutralization assay using monoclonal antibodies in different target cells also support the phenomenon, with antibodies targeting epitopes outside the receptor binding motif showing reduced potency in cells with higher ACE2 expression. The manuscript was well-written, with clear structure and consistent logic, and the conclusions were solid based on the experimental results. Although unable to directly compare the ACE2 expression level of the constructed target cells with those in the real biological cases, the conclusion from this manuscript will provide an important factor for consideration when evaluating neutralizing antibodies in the future using different neutralization assay platforms. I would recommend acceptance of this manuscript, with several minor points addressed as follows.
In Figure 1B, the authors revealed that different ACE2 expression level in the constructed target cells can largely affect the infectivity of the pseudoviruses. Therefore, I’m wondering how the authors normalized the neutralization results with different virus titers? And how would the different virus infectivity levels may affect the neutralization results in terms of saturation, incomplete neutralization, etc.?
The authors utilized antibody depletion method to eliminate the RBD-specific polyclonal antibodies from human serum samples, to reveal the contribution of RBD antibodies to the total neutralization. Would it be possible if the authors could deplete antibodies targeting other epitopes other than RBD as a more complete comparison?
The authors assessed the neutralization abilities of three monoclonal antibodies targeting different epitopes on SARS-CoV-2, including Ly-CoV555 targeting receptor-binding motif of the RBD, S309 targeting the RBD outside receptor binding motif, and 4A8 targeting NTD. However, they have different levels of potencies, especially S309 has ~10-fold weaker potency. It may be better if the authors could include more monoclonal antibodies for a better comparison? In addition, besides RBD and NTD, there are monoclonal antibodies targeting the S2 epitope, maybe the authors can also take these S2 antibodies into account for comparison.
It is known that besides ACE2, TMPRSS2 is another important factor during SARS-CoV-2 virus entry into target cells. Did the authors look into the expression levels of TMPRSS2 in the target cells in addition to ACE2 itself?
In Figure 2D, the authors may change the y-axis from ranging 0-100 to ranging 50-100, so that the difference between groups may be more obvious to the readers.
Author Response
Thanks for the favorable comments. To address the specific questions:
- We used the same virus stock and dilution for all cells, ensuring we had >500K RLUs per well to have adequate signal.
- It would be interesting to deplete sera to other epitopes, however we do not have the correct magnetically labeled protein domains to do this. Also, since the RBD is the dominant target of neutralization (>50% in all cases), removing other antibodies will have minimal effects on overall neutralization unless RBD antibodies are removed first.
- We agree it could be interesting to look at more antibodies. However, the major focus of our study is how ACE2 expression affects overall serum neutralization against RBD versus non-RBD antibodies. Therefore, also trying to examine S2 antibodies is outside the scope of our current study although we agree it is an interesting area for future work.
- We agree it would also be interesting to look at TMPRSS2 expression, but we do not have isogenic 293T cells expressing different levels of TMPRSS2 so this is outside the scope of our current work and would require another study.
- We prefer to keep the minimum at 0 to emphasize that in all situations the majority of the neutralization is due to RBD antibodies. We think it's still possible to see differences among groups on this scale.